# Evidence for spin current driven Bose-Einstein condensation of magnons

B. Divinskiy[1,4], H. Merbouche [2,4], V. E. Demidov [1✉], K. O. Nikolaev[1], L. Soumah[2], D. Gouéré[2], R. Lebrun [2], V. Cros [2], Jamal Ben Youssef[3], P. Bortolotti[2], A. Anane[2] & S. O. Demokritov[1]

The quanta of magnetic excitations – magnons – are known for their unique ability to undergo Bose-Einstein condensation at room temperature. This fascinating phenomenon reveals itself as a spontaneous formation of a coherent state under the influence of incoherent stimuli. Spin currents have been predicted to offer electronic control of Bose-Einstein condensates, but this phenomenon has not been experimentally evidenced up to now. Here we show that current-driven Bose-Einstein condensation can be achieved in nanometer-thick films of magnetic insulators with tailored nonlinearities and minimized magnon interactions. We demonstrate that, above a certain threshold, magnons injected by the spin current over-populate the lowest-energy level forming a highly coherent spatially extended state. We quantify the chemical potential of the driven magnon gas and show that, at the critical current, it reaches the energy of the lowest magnon level. Our results pave the way for implementation of integrated microscopic quantum magnonic and spintronic devices.

[1] Institute for Applied Physics, University of Muenster, Corrensstrasse 2-4, 48149 Muenster, Germany. [2] Unité Mixte de Physique, CNRS, Thales, Université Paris-Saclay, 91767 Palaiseau, France. [3] LABSTICC, UMR 6285 CNRS, Université de Bretagne Occidentale, 29238 Brest, France. [4] These authors contributed equally: B. Divinskiy, H. Merbouche ✉email: demidov@uni-muenster.de

For a long time, the only experimentally proven approach enabling observation of room-temperature Bose–Einstein condensation (BEC) of magnons was the approach based on the utilization of microwave pumping[1–10]. This approach relies on the injection of low-energy non-equilibrium magnons within a relatively small interval of energies, followed by their thermalization and accumulation in the lowest-energy level. Although this process was shown to result in a formation of the quasi-equilibrium magnon gas, which can be described by a non-zero chemical potential and an effective temperature, it was found to predominantly affect low-energy magnons, which do not fully equilibrate with high-energy ones resulting in a very high effective temperature of the former[5]. Another recently demonstrated approach relies on the dynamic spectral redistribution of magnons in an overheated magnon gas caused by its rapid cooling[11,12]. Due to its dynamic origin, this approach can only be used to create BEC-like states for a few tens of nanoseconds.

An alternative way to drive magnon gas into a steady quasi-equilibrium state characterized by a non-zero chemical potential[13–16] can be based on the utilization of dc electric currents converted into pure spin currents by the spin-Hall effect (SHE)[17,18]. In strong contrast to microwave pumping, the spin torque exerted by the spin current simultaneously affects the population of all magnon modes[19,20]. Importantly, it hence allows avoiding strongly non-equilibrium transient states of the magnon gas. The possibility to create magnon BEC by using the above driving mechanism has been theoretically predicted a long time ago[21–23]. However, up to now, no direct experimental confirmation of the current-driven BEC has been reported. Here, we show that this mechanism can indeed lead to the formation of stationary equilibrium magnon BEC, which has not been achieved by using previously demonstrated approaches.

The most striking and necessary manifestation of the magnon BEC is the spontaneous formation of a coherent dynamical state. However, the observation of such a state alone is insufficient to claim the BEC transition[24]. Indeed, every magnetic oscillator driven by spin-transfer effects[25–32] exhibits more or less coherent magnetic oscillations. Although these oscillations are formed spontaneously, the underlying mechanism is not necessarily BEC. Instead, it can be a strongly non-equilibrium process, similar to that responsible for the generation of coherent optical radiation in lasers. Strictly speaking, an assertion of unambiguous BEC identification requires that the magnon gas remains in a (quasi-)equilibrium thermodynamic state: while the emergent coherent dynamics is described in terms of the overpopulation of the lowest-energy magnon level, the population of the rest of the magnon states follows the Bose–Einstein statistics. In addition, the formation of coherent magnetic dynamics via BEC, which is the condensation of particles in the phase space, is inconsistent with the formation of a dynamical bullet[27,33] or droplet[34–36], which represents the condensation of particles in the real space due to their attractive interaction. As a general rule of thumb, the nonlinear interactions between magnons at the lowest-energy level is one of the main factors hindering the magnon BEC[15].

In relatively thick films of yttrium iron garnet (YIG), where microwave-driven BEC has been evidenced[1–10], the magnons at the lowest-energy level exhibit very weak nonlinear interactions resulting in a stabilization of BEC by extrinsic effects[10,37]. In contrast, in nanometer-thick YIG films, which can be efficiently driven by spin currents, the energy of the lowest magnon level contains a significant dynamic dipolar contribution, which results in a strong attractive interaction between magnons leading to spatial instabilities and collapses[27,38]. The nonlinearities associated with dipolar effects have also been recently shown to be responsible for the strong scattering of magnons out from the lowest-energy state preventing its overpopulation[39].

Here, we study a current-driven system based on a nanometer-thick Bi-substituted YIG film, in which the out-of-plane magnetic anisotropy has been engineered in order to fully compensate the dipolar demagnetizing field resulting in vanishing nonlinear interactions of the lowest-energy magnons. We show that, in such a system, the gas of weakly interacting magnons can undergo BEC under the influence of pure spin currents. The BEC transition is documented by the observation of the overpopulation of the lowest-energy level resulting in a spatially extended dynamical state exhibiting high temporal and spatial coherence. Additional confirmation of the BEC nature of the observed transition is provided by the measurements of the current-dependent density of high-energy magnons. In agreement with the BEC scenario, this density saturates, as the chemical potential of the magnon gas reaches the lowest magnon energy, while the density of lowest-energy magnons continues to grow. This observation indicates that the additionally injected magnons accumulate in the lowest-energy level, which is a clear signature of the BEC transition.

## Results

**Studied system and experimental approach.** In Fig. 1a, we show the schematics of our experiment. The studied system is based on a 20-nm thick film of Bi-doped (one Bi atom per chemical unit) BiYIG ($Bi_1Y_2Fe_5O_{12}$) grown by the PLD on substituted gallium gadolinium garnet (sGGG) substrate[40]. The film is magnetized to saturation by a static in-plane magnetic field $H_0$. It exhibits significant strain-induced perpendicular magnetic anisotropy (PMA) with the effective anisotropy field $\mu_0 H_a = 167$ mT, which is very close to the saturation magnetization of the film $\mu_0 M_s = 175$ mT. This leads to the compensation of the effects associated with the dipolar anisotropy and thus results in the minimization of nonlinear magnon interactions[39]. The spin system of the BiYIG film is driven out of the equilibrium by spin torque through the injection of the pure spin current $I_s$ created by the SHE in the Pt electrode carrying dc electric current $I$. Depending on the polarity of the current, the injected spin current either increases or decreases the magnon population within a broad interval of energies[19] resulting in the formation of the new quasi-equilibrium state of the magnon gas, which can be described by the Bose–Einstein distribution with a non-zero chemical potential[13,15].

We experimentally evaluate the population of the magnon states by using the micro-focus Brillouin light scattering (BLS) spectroscopy[41] (see Methods for details). The measured signal—the BLS intensity—is proportional to the spectral density of magnons at the position, where the probing laser light is focused (Fig. 1a). By moving the focal spot over the surface of the BiYIG film, we additionally obtain information about the spatial variations of the spectral distribution of magnons.

Figure 1b shows the BLS spectrum recorded at $\mu_0 H_0 = 100$ mT and $I = 0$, describing magnons, which exist in the magnetic film due to thermal fluctuations at room temperature. Under these conditions, the magnon gas is at thermal equilibrium with the lattice and is characterized by zero chemical potential and room temperature[1]. The recorded spectrum exhibits an asymmetric peak with the maximum at 2.9 GHz with the high-frequency tail extending to about 3.3 GHz and an additional nearly symmetrical peak located at 39 GHz.

Figure 1c shows the dispersion spectrum of magnons calculated according to ref. [42], which allows identification of the peaks. The low-frequency part of the spectrum corresponds to magnons characterized by a nearly uniform distribution of the dynamic magnetization across the film thickness (see the inset in Fig. 1c). These magnons exhibit an anisotropic dispersion in the plane of the film, as seen from the essentially different dispersion

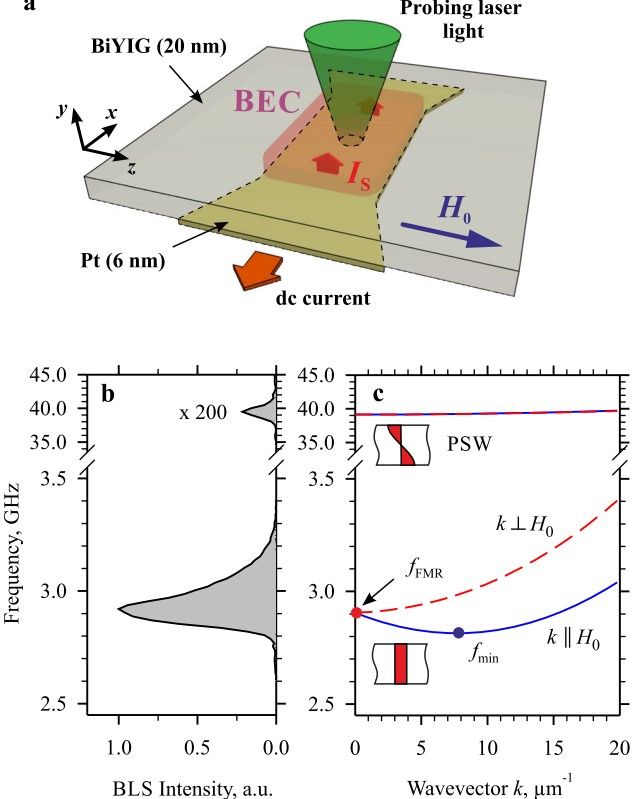

**Fig. 1 Experimental system. a** Schematics of the experiment. Magnon gas in a BiYIG film with PMA is driven by the injection of the spin current $I_s$ created by SHE in the Pt electrode with an active area of $1 \times 2\ \mu m^2$. The spectral distribution of magnons is measured by analyzing the light inelastically scattered from magnons. Transparent sGGG substrate is not shown for clarity. **b** Normalized BLS spectrum of magnons in the BiYIG film measured at $I = 0$ and $\mu_0 H_0 = 100$ mT. **c** Calculated dispersion spectrum of magnons in the BiYIG film. Solid curves correspond to magnons with the wavevector $k$ oriented parallel to $H_0$. Dashed curves correspond to magnons with the wavevector $k$ oriented perpendicular to $H_0$. $f_{FMR}$ marks the frequency of the ferromagnetic resonance. $f_{min}$ marks the frequency of the lowest-energy magnon state. Insets schematically show the distributions of the dynamic magnetization across the film thickness for the fundamental and the PSW magnon mode.

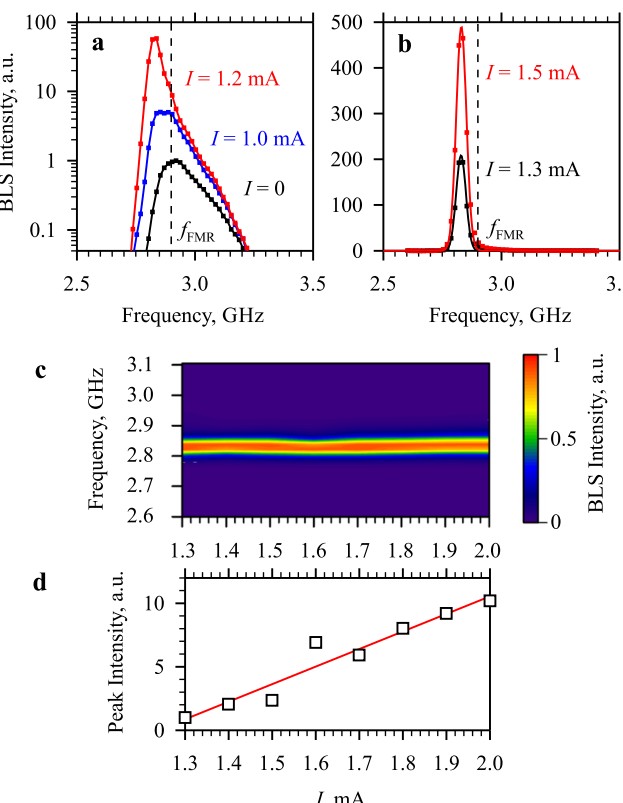

**Fig. 2 Formation of the Bose–Einstein condensate of magnons. a**, **b** Representative BLS spectra recorded at different currents in the Pt electrode, as labeled. Dashed lines mark the frequency of the ferromagnetic resonance. Note formation of the narrow intense spectral peak at $I = 1.3$ mA. **c**, **d** Modifications of the formed peak at $I \geq 1.3$ mA. **c** Color-coded normalized BLS spectra in the current–frequency coordinates. **d** Current dependence of the peak intensity normalized by its value at $I = 1.3$ mA. Solid line—a guide for the eye. The data were obtained at $\mu_0 H_0 = 100$ mT.

curves corresponding to magnons with the wavevector $k$ oriented parallel (solid curves) and perpendicular (dashed curves) to the direction of the static field. The two dispersion curves merge at $k = 0$ at the frequency of the uniform ferromagnetic resonance $f_{FMR}$. The dispersion curve characterizing magnons with $k \perp H_0$ shows a monotonous rise, while that for magnons with $k \parallel H_0$ shows non-monotonous behavior resulting in the appearance of a minimum at $k = 7.6\ \mu m^{-1}$ and the frequency $f_{min}$, which is by 0.09 GHz smaller than $f_{FMR}$. The state corresponding to this minimum is the lowest-energy magnon state, where BEC is expected to take place.

As seen from the comparison of the data of Fig. 1b, c, the experimentally observed low-frequency peak has the maximum at $f_{FMR}$. This is due to the wavevector-dependent sensitivity of the BLS technique, which maximizes at $k = 0$ and gradually decreases with the increase of $k$. Accordingly, the BLS intensity decreases with the increase of the frequency and vanishes at frequencies corresponding to $k > k_{max}$, which is mainly determined by the wavevector of the probing light. Note here that, at $I = 0$, the BLS intensity at $f_{min}$ is significantly smaller than that at $f_{FMR}$ due to the significantly smaller experimental sensitivity.

The small high-frequency BLS peak in Fig. 1b corresponds to the so-called perpendicular standing-wave (PSW) magnon modes[11,43,44], which possess a non-uniform distribution of the dynamic magnetization across the film thickness (see the inset in Fig. 1c). These modes are characterized by a nearly isotropic dispersion ($k \parallel H_0$ and $k \perp H_0$ curves are indistinguishable in Fig. 1c). As will be shown below, although these modes are located at frequencies far above the frequency of the lowest-energy magnon state, one can use the possibility to measure their population to prove that the spin current enhances the population of magnon states in a broad frequency range and that the evolution of the magnon gas follows the BEC scenario.

**Formation and characteristics of magnon BEC.** We now analyze the modifications of the magnon spectral distribution caused by the injection of the spin current. In agreement with the symmetry of the SHE, at $I < 0$, the magnon density is found to monotonically decrease with the increase of the magnitude of the current, while, at $I > 0$, the magnon density exhibits a strong enhancement (see Supplementary Fig. 1). In the following, we focus on the latter process, as it is expected to cause BEC at sufficiently large $I$. As seen from the data shown in Fig. 2a, at small currents, the population of all magnon states becomes increasingly enhanced with the maximum effect observed for states at lower frequencies. In particular, the shape of the spectrum recorded at $I = 1.2$ mA clearly indicates that the population

of the states at frequencies close to $f_{min}$ grows much faster in comparison with that at $f_{FMR}$, as expected for the chemical potential gradually approaching the minimum energy. Finally, at $I > 1.3$ mA, a narrow intense peak appears in the spectra (Fig. 2b) at the frequency, which is by about 0.08 GHz below $f_{FMR}$. This frequency difference is very close to the theoretically estimated value $f_{FMR} - f_{min} = 0.09$ GHz. This allows us to conclude that the observed peak corresponds to the overpopulated lowest-energy magnon state, which is a first indication of the onset of BEC in the driven magnon gas.

We emphasize that the frequency of the peak and its spectral width remain nearly constant within the entire range $I = 1.3–2.0$ mA (see Fig. 2c, which shows the normalized color-coded BLS spectra in the current-frequency coordinates), while the peak intensity increases by more than one order of magnitude (Fig. 2d). Note that, in magnetic films without PMA, the frequency is known to exhibit a strong negative shift with the increase of the density of magnons[33,45]. Generally, this means that the local increase of the particle density is energetically favorable, which corresponds to the attractive interaction between them causing local accumulation of particles in the real space followed by a collapse—the phenomenon, which is known to hinder the formation of stable BEC[46,47]. As follows from the independence of the frequency of the observed peak from its intensity (Fig. 2c, d), thanks to the effects of PMA, the attractive interactions are well suppressed in our system, enabling the BEC transition.

We also note that the Oersted field of the current in the Pt electrode has a negligible effect on the frequency of the observed peak. For the used geometry, the variation of the total static magnetic field with the current can be estimated as 0.6 mT mA$^{-1}$. Calculations of the dispersion spectrum show that this corresponds to the frequency shift of about 0.01 GHz over the range $I = 1.3–2.0$ mA, where the BEC peak is observed. This shift is smaller than the frequency resolution of the BLS apparatus and cannot be seen in the data of Fig. 2c.

We then address the spectral coherence of the observed BEC. Because of the limited frequency resolution of the BLS apparatus, it is not possible to directly measure the spectral linewidth of the formed monochromatic oscillations. The fitting of the BLS spectra shown in Fig. 2b with the Gaussian function yields the full width at half a maximum of 0.051 GHz. This value is very close to the frequency resolution of the BLS apparatus, which, for the used experimental arrangement, can be estimated as 0.04–0.05 GHz. Therefore, we can conclude that the spontaneously formed dynamical state is characterized by a high temporal coherence. To illustrate that the state is also coherent in the space domain, we perform spectral measurements at different spatial locations within the active area of the electrode. Figure 3a, b shows the color-coded BLS spectra recorded by moving the probing spot along the z- and x-direction, as schematically shown in the corresponding insets. As seen from these data, the frequency and the linewidth of the BLS peak do not change noticeably across the entire $1 \times 2$ μm$^2$-large active area, indicating that a spatially extended BEC is formed in the studied system.

Note here that, in contrast to most of the previously studied systems exhibiting current-driven magnetic oscillations[25–32], our system does not show transitions to multi-frequency oscillations or mode jumps—behaviors, which are inconsistent with the BEC scenario, where all additional magnons created at large driving stimuli should accumulate in the single, lowest-energy state. The latter condition can be unambiguously proven by analyzing the current-dependent population of a spectral state corresponding to the magnon energy far above the energy of the lowest state. To perform this analysis, we utilize the possibility to experimentally access PSW magnons at the frequency 39 GHz (Fig. 1b). In Fig. 4a, we show the current-dependent intensity of the PSW peak

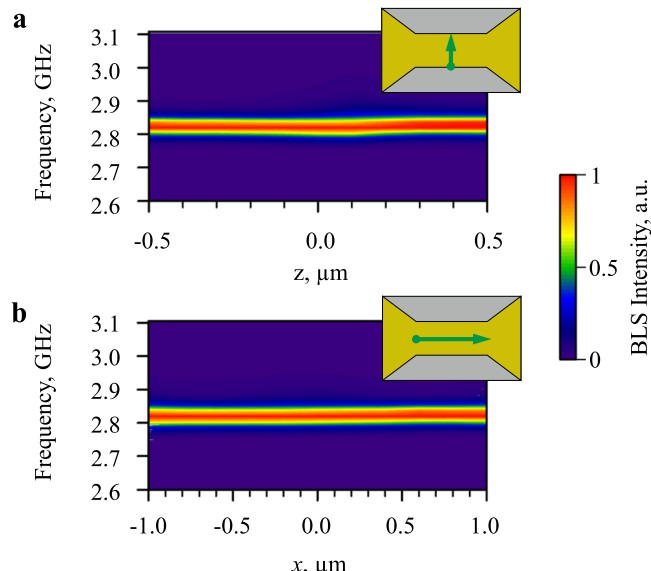

**Fig. 3 Spatial characterization of the BEC state. a, b** Color-coded normalized BLS spectra recorded by moving the probing spot along the z- and x-direction, as schematically shown in the corresponding insets. The data were obtained at $I = 1.3$ mA and $\mu_0 H_0 = 100$ mT.

together with the same dependence for the lowest magnon state at $f_{min}$. First, the data show that the small-current enhancement of the population of the high-frequency PSW states is significantly smaller than that of the state at $f_{min}$ (note different vertical scales). At $I = 1$ mA, the enhancement of the BLS intensity of the PSW peak does not exceed 20%, while the intensity at $f_{min}$ increases by nearly a factor of 10. This result is consistent with the assertion that the main effect of the spin current on the magnon gas is the increase of its chemical potential[13–16]. More importantly, at currents where the formation of the coherent state takes place (Fig. 2b), the intensity of the PSW peak saturates, while the BLS intensity at $f_{min}$ exhibits further strong growth. This result indicates that, in agreement with the BEC scenario, additional magnons created at large currents do not spread over the entire energy space, but instead overpopulate the lowest-energy state.

A key parameter to characterize the BEC transition is the chemical potential that we extract from the acquired data following the approach in Ref. [11]. As seen from Fig. 4b, the chemical potential expressed in frequency units $\mu h^{-1}$ (here $h$ is the Plank constant) monotonically increases at small currents, reaches $f_{min}$ at $I = I_C = 1.3$ mA, where the formation of the narrow spectral peak is observed (Fig. 2b), and then saturates. This feature allows us to clearly identify the BEC transition.

In order to determine the characteristic temporal scale of the BEC onset, we perform additional time-resolved measurements. We apply electric current in the form of 1 μs long pulses and record the BLS intensity as a function of delay with respect to the start of the current pulse. Figure 4c shows the temporal dependence of the BLS intensity at the frequency of the BEC peak recorded at the maximum used current of 2 mA. The data of Fig. 4c show that the intensity of the BEC peak first increases exponentially (note the logarithmic scale) and then saturates after 100–150 ns. This indicates that the discussed driving mechanism is relatively fast and can be used for studies of dynamical BEC phenomena.

**Dependence on the static magnetic field.** Finally, we aim at determining the conditions of the BEC formation over a broad

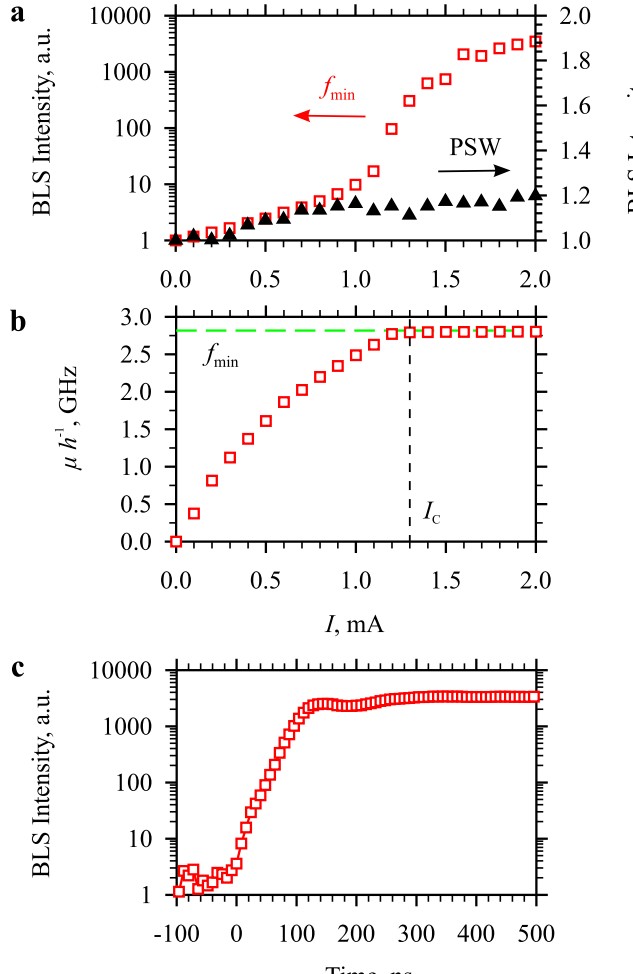

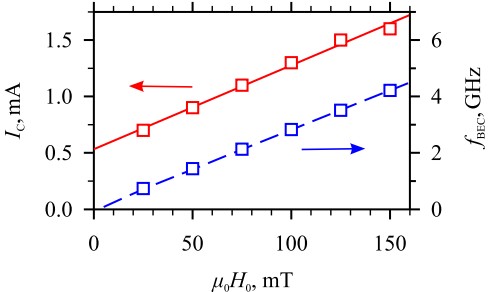

**Fig. 5 Dependence on the magnetic field.** Static-field dependence of the critical current $I_C$ and of the BEC frequency $f_{BEC}$. A solid line is the linear fit of the experimental data. Dashed curve—calculated frequency of the lowest-energy magnon state $f_{min}$.

sum of two terms: the field-independent term $\omega_{r0}$ originating from the influence of the inhomogeneity of the magnetic film, and the term $\alpha\gamma H_0$, which scales linearly with $H_0$ (here $\alpha$ is the Gilbert damping constant and $\gamma$ is the gyromagnetic ratio)[29]. The found almost linear field dependence of the critical current shows that this simple description is applicable within the used range of the static field.

In conclusion, our experimental results provide the direct experimental evidence of room-temperature BEC in a magnon gas driven by spin currents—the phenomenon, which was theoretically predicted nearly a decade ago and has been actively discussed in the scientific community since then. We show that stable spatially extended BEC of magnons can be created in systems, where nonlinear magnon interactions are minimized by using compensation of the dipolar effects by the anisotropy. These findings open new avenues for the studies of magnon BEC by demonstrating an approach enabling the creation of magnon BEC without breaking the equilibrium between low-energy and high-energy magnons. The simple and robust driving method based on the utilization of SHE at the micrometer scale enables the implementation of a large variety of new experiments. We believe that our findings will strongly move forward the field of magnon thermodynamics and quantum magnetic phenomena in general.

## Methods

**Sample fabrication.** The growth of BiYIG films by pulsed laser deposition (PLD) is realized using a stoichiometric BiYIG target. The laser used is a frequency tripled Nd:YAG laser ($\lambda = 355$ nm), of a 2.5 Hz repetition rate and a fluency of about 1 J cm$^{-2}$, and a substrate temperature of about 500 °C. The distance between the target and the substrate is fixed at 44 mm. Prior to the deposition, the substrate is annealed at 700 °C under 0.4 mbar of $O_2$. For the growth, the pressure is set at 0.25 mbar $O_2$ pressure. At the end of the growth, the sample is cooled down under 300 mbar of $O_2$. No post annealing is performed. Deposition of Pt is performed using dc magnetron sputtering. Prior to Pt deposition, a slight $O_2$ etch is performed to remove photo-resist residues and promote surface spin transparency. The Pt electrode is 6 nm thick, its width is reduced to 1 μm over a length of 2 μm. Hence, the current density and the resulting spin–orbit torque is maximum over this $1 \times 2$ μm$^2$ constriction.

*Micro-focus BLS measurements.* All the measurements were performed at room temperature. The measurements of current-dependent magnon spectral distributions were based on the analysis of the inelastic scattering of laser light from magnons. The probing light with the wavelength of 532 nm and the power of 0.1 mW was produced by a single-frequency laser possessing the spectral linewidth <10 MHz. The light was focused through the sample substrate (sGGG) into a submicrometer-size diffraction-limited spot by using a 100× corrected microscope objective lens with the numerical aperture of 0.85. The scattered light was collected by the same lens and analyzed by a six-pass Fabry–Perot interferometer. The lateral position of the probing spot was controlled by using a high-resolution custom-designed optical microscope and was actively stabilized with a precision better than 50 nm.

**Fig. 4 Thermodynamic and temporal characteristics of the current-driven magnon gas. a** Current dependences of the BLS intensity for the PSW peak and that for the lowest magnon state at $f_{min}$, as labeled. The data are normalized by the value at $I = 0$. Note different vertical scales for the two dependencies. The data were obtained at $\mu_0H_0 = 100$ mT. **b** Current dependence of the chemical potential expressed in frequency units. Horizontal dashed line marks the frequency of the lowest-energy magnon state $f_{min}$. A vertical dashed line marks the critical current $I_C$, at which the formation of the BEC peak is observed. **c** Temporal dependence of the BLS intensity at $f_{min}$ obtained at $I = 2$ mA and $\mu_0H_0 = 100$ mT.

range of the static magnetic field $H_0$. The experimental data show that the variation of the static field mainly influences the frequency of the condensate and the value of the critical current $I_C$, while the scenario of the BEC formation remains qualitatively the same (compare the data in Fig. 2a, b with those presented in Supplementary Fig. 2). As seen from the data of Fig. 5, both the BEC frequency and $I_C$ exhibit nearly linear dependence on $H_0$. Within the entire addressed field range, the experimentally determined BEC frequency matches exactly the calculated frequency of the lowest-energy magnon state $f_{min}$ (dashed curve in Fig. 5). The linear variation of the BEC frequency with $H_0$ indicates that the dipolar demagnetizing fields are precisely compensated by the anisotropy. The linear field dependence of the critical current extrapolates to a finite value at $H_0 = 0$. This result is expectable since the efficiency of the magnon creation by the spin torque depends on the magnon relaxation frequency $\omega_r$. For our system, where the dipolar fields are compensated by the anisotropy, in the first approximation, $\omega_r$ can be considered as a

## Data availability

The data that support the findings of this study are available in Zenodo with the identifier https://doi.org/10.5281/zenodo.5564234.

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

## Acknowledgements

This work was supported in part by the Deutsche Forschungsgemeinschaft (Project number 416727653) and by the ANR MAESTRO project, Grant no. 18-CE24-0021 of the French Agence Nationale de la Recherche. We also acknowledge financial support from the Horizon 2020 Framework Program of the European Commission under FET-Open grant agreement no. 899646 (k-NET).

## Author contributions

B.D., H.M. and K.O.N. performed measurements and data analysis. L.S., D.G. and J.B.Y. grew and characterized the films. H.M. performed the nanofabrication. R.L., V.C. and P.B. contributed to the design and implementation of the research. V.E.D., A.A. and S.O.D. formulated the experimental approach and supervised the project. All authors co-wrote the paper.

## Funding

## Competing interests

The authors declare no competing interests.
