## [Peer Review File · Nature Communications]

Reviewers' Comments:

Reviewer #1:

Remarks to the Author:

The manuscript reports on the experimental observation of the Bose-Einstein condensation (BEC) of magnons induced by the spin-Hall effect. The subject is interesting and the manuscript itself is well written. It definitely deserves publication in some form, but I cannot recommend it for publication in the Nature Communications.

The manuscript represents a continuation of efforts of S.O. Demokritov's group on studying the BEC of magnons at room temperature in YIG films using the BLS spectroscopy. Their initial, in 2006, Nature publication on experimental observation of the room temperature BEC of magnons was very interesting and inspiring and in the very recent Nature Communications paper they published the experimental evidence of spatial stability of magnons' condensate. Their present work is also devoted to observation of the room-temperature BEC of magnons, but now the authors report on the observation of the spin-current driven condensation. This subject is well understood theoretically and there are other experimental groups, demonstrating similar results (for example, "Control of the Bose-Einstein Condensation of Magnons by the Spin-Hall Effect", arXiv:2102.13481v1). I do not think that at these conditions another experimental observation of the room-temperature BEC of magnons, even not driven by microwave pumping as before, requires publication at the Nature level.

To conclude, I cannot recommend publication of this manuscript in the Nature Communications. Instead, I would recommend to transfer it to the Communication Physics, where the authors could present more experimental details, important for groups working in the field of magnons' BEC.

Reviewer #2:

Remarks to the Author:

The article "Evidence for spin current driven Bose-Einstein condensation of magnons" by Divinskiy et al. reports on the observation of a magnon Bose-Einstein condensate formed by a spin current injection in a Bi-doped YIG. The observation has been made by means of micro-focused Brillouin light scattering technique.

The article is well written and demonstrates results, which could be published in Nature Communications after a revision. There are several details and evidence, which must be presented to fully support the conclusions made by Authors.

On line 87 Authors describe the composition of the sample. It is described as Bi-doped Yttrium Iron Garnet. Is the concentration of Bismuth in the sample exactly one atom per chemical unit of YIG?

The measurements show that there is no shift in the spectral position of the observed signals upon application of the current. However, there should be an influence from the magnetic field generated by the electric current. The proximity of the conductor and huge current density should lead to a significant magnetic field. The estimates or direct measurements have to be added to the paper to exclude that effect.

In the manuscript, Authors present results only for one direction of the current through the Pt layer. However, to fully provide evidence of spin Hall effect driven injection, the results of opposite current polarity have to be presented. Otherwise, it is unclear whether the effect is not caused by the Spin Seebeck effect. The evidence must be presented and the corresponding discussion has to be added to the manuscript.

Could authors comment on the time scale of onset of the BEC in the observed current-driven scenario? Such discussion could greatly help the reader to understand the dynamics of the process.

Concluding my report, I cannot recommend publication in its current form. However, I would be happy to reconsider my decision if the required evidence will be added to the manuscript.

Reviewer #3:

Remarks to the Author:

Report on manuscript NCOMMS-21-14513-T

Title: Evidence for spin current driven Bose-Einstein condensation of magnons

Authors: B. Divinskiy et al

This paper reports very interesting experiments showing evidences that it is possible to produce Bose-Einstein condensation (BEC) of magnons in a film of yttrium iron garnet (YIG) by spin currents generated by the conversion of dc electric currents through the inverse spin Hall effect (ISHE) in an adjacent platinum layer. In previous experiments on the formation of magnon BEC, the magnon system is excited by means of parametric pumping with pulsed microwave driving or rapid cooling by pulsed heating. In all experiments, the redistribution of the magnon system leading to the BEC is observed with Brillouin light scattering (BLS). In both cases the excitation of the magnon system raises the chemical potential, leading to the BEC when this reaches the minimum energy of the magnon dispersion relation in the YIG film, so in both processes the BEC state is observed by BLS only in a short time interval. Here, on the contrary, the excitation of the magnon system is made by a dc current, and the features of the BEC are observed with BLS in an equilibrium situation as the current intensity is raised above a critical value. In my opinion, the results presented in this manuscript constitute an important contribution to Bose-Einstein condensation phenomena which would be of interest to condensed matter physicists as well as to a broader community of scientists, and thus the paper could be published in Nature Communications. However, before acceptance of the manuscript, the authors would have to clarify some points and add some figures. The following specific points must be addressed:

1- In lines 61-63, it is stated that "As a general rule of thumb, the nonlinear interactions between magnons is one of the main factors hindering the magnon BEC." However, as I understand, Ref. 3 shows that the magnon interactions are crucial for the magnon redistribution and also for the quantum coherence of the states with frequencies near the minimum, which is a condition for a Bose-Einstein condensate, as shown in Ref. 23. Or am I wrong about the role of the magnon interactions?

2- The authors state that the peak at 39 GHz corresponds to a perpendicular standing-wave (PSW) magnon mode and cite Refs 42,43. However, in both references the detected magnon modes have much smaller frequencies. Is there another reference that shows a quantitative estimate of the frequency of the PSW magnon for a 20 nm thick YIG film?

3- If I understand correctly, the mechanism proposed for the formation of the BEC is the redistribution of the magnetic energy pumped into the PSW magnon by the spin current. Why couldn't the spin current excite magnons in a broad frequency and wave vector range, and these would then condense at the state with minimum frequency.

4- A crucial test for any experiment evidencing a new phenomenon involving magnons is the dependence on the magnetic field intensity. Figure 5 shows the field dependence of the critical current for BEC formation. The authors argue that the linear field dependence arises from the fact that the magnon relaxation rate varies linearly with frequency. However, this is true for magnons with small wave numbers, such as the one with minimum frequency. If I understand correctly, the magnons driven by the spin current are the ones in the PSW mode, and certainly these do not have relaxation rates that vary linearly with field.

5- Again, a crucial test for any experiment evidencing a new phenomenon involving magnons is the dependence on the magnetic field intensity. It is important that the authors include figures like Figs. 2a and 2b, showing the BLS spectra for another field value, say 200 mT, and one like Fig. 5 showing the field dependence of the BEC frequency.

Response to Reviewer #1

The Reviewer writes:

The manuscript reports on the experimental observation of the Bose-Einstein condensation (BEC) of magnons induced by the spin-Hall effect. The subject is interesting and the manuscript itself is well written. It definitely deserves publication in some form, but I cannot recommend it for publication in the Nature Communications.

The manuscript represents a continuation of efforts of S.O. Demokritov's group on studying the BEC of magnons at room temperature in YIG films using the BLS spectroscopy. Their initial, in 2006, Nature publication on experimental observation of the room temperature BEC of magnons was very interesting and inspiring and in the very recent Nature Communications paper they published the experimental evidence of spatial stability of magnons' condensate.

Reply:

We thank the Reviewer for the appreciation of the importance of our previous works on the room-temperature magnon BEC. We would like to point out that the referees of our initial Nature paper also expressed concerns about the significance of our findings, but were finally convinced by our arguments. We hope that, the arguments given below will convince the Reviewer that our present work also represents a major step forward in the field and deserves publication in Nature Communications.

The Reviewer writes:

Their present work is also devoted to observation of the room-temperature BEC of magnons, but now the authors report on the observation of the spin-current driven condensation. This subject is well understood theoretically and there are other experimental groups, demonstrating similar results (for example, "Control of the Bose-Einstein Condensation of Magnons by the Spin-Hall Effect", arXiv:2102.13481v1).

Reply:

We respectfully disagree with the Reviewer that "this subject is well understood theoretically and there are other experimental groups, demonstrating similar results".

We emphasize that the theoretical works predict the possibility of the current-driven BEC in systems, where it could not be observed experimentally for many years. As was shown in several publications (see, e.g., Nat. Commun. 8, 1579 (2017)), in real nanometer-thick magnetic films, the formation of BEC is hindered by the magnon-magnon interactions, which is not taken into account in the existing theoretical models. This fact already indicates an incomplete theoretical understanding. This is the main reason why it took so many years to observe this theoretically predicted phenomenon in a real experiment. After many-years research, we were finally able to find an approach, which allows one to minimize the adverse nonlinear interactions. In our work, we provide an experimental evidence that this approach allows one to achieve current-driven BEC.

Our work is the first and only experimental demonstration of the current-induced magnon BEC. The paper mentioned by the Reviewer does not report similar results. In this paper, which is the extension of the work Ref. 11, a short-living BEC-like state was excited by the strongly non-equilibrium rapid-cooling process, whereas the weak spin-Hall effect could only result in a few-percent variation of the onset threshold.

To make the latter fact clearer for the readers, in the revised manuscript, we have cited the mentioned paper together with Ref. 11.

The Reviewer writes:

I do not think that at these conditions another experimental observation of the room-temperature BEC of magnons, even not driven by microwave pumping as before, requires publication at the Nature level.

Reply:

We would like to emphasize that our work does not simply demonstrate “another experimental observation of the room-temperature BEC of magnons, even not driven by microwave pumping as before”. In contrast to microwave pumping, the demonstrated approach allows one to create BEC without breaking the equilibrium between low-energy and high-energy magnons, which is known to result in extreme effective temperatures of the microwave-driven magnon gas (up to 5000 K). In addition, unlike the rapid-cooling process, our approach allows one to maintain stable state of the magnon condensate for unlimited time. Moreover, due to its simplicity and robustness, our approach enables implementation of a large variety of exciting new experiments and, therefore, opens new avenues for studies of magnon BEC.

We also would like to note, that the field of room-temperature magnon BEC remains attractive to a wide scientific audience, which is confirmed by numerous publications at the Nature level. Let us mention Nat. Nanotechnol. 15, 457 (2020) as the most recent one.

The Reviewer writes:

To conclude, I cannot recommend publication of this manuscript in the Nature Communications. Instead, I would recommend to transfer it to the Communication Physics, where the authors could present more experimental details, important for groups working in the field of magnons' BEC.

Reply:

We hope that the Reviewer will find our arguments convincing and will recommend publication of the revised manuscript in Nature Communications.

Response to Reviewer #2

The Reviewer writes:

The article "Evidence for spin current driven Bose-Einstein condensation of magnons" by Divinskiy et al. reports on the observation of a magnon Bose-Einstein condensate formed by a spin current injection in a Bi-doped YIG. The observation has been made by means of micro-focused Brillouin light scattering technique.

The article is well written and demonstrates results, which could be published in Nature Communications after a revision. There are several details and evidence, which must be presented to fully support the conclusions made by Authors.

Reply:

We thank the Reviewer for the positive evaluation of our work and for the constructive comments aimed at the improvement of our manuscript. Below, we respond in detail to all the Reviewer's questions and inquiries, and describe how they have been addressed in the revised manuscript. We hope that the Reviewer will find our responses and revisions satisfactory and will recommend publication of the revised manuscript.

The Reviewer writes:

On line 87 Authors describe the composition of the sample. It is described as Bi-doped Yttrium Iron Garnet. Is the concentration of Bismuth in the sample exactly one atom per chemical unit of YIG?

Reply:

This is correct. Addressing the Reviewer's question, we have clarified this on the page 4 of the revised manuscript.

The Reviewer writes:

The measurements show that there is no shift in the spectral position of the observed signals upon application of the current. However, there should be an influence from the magnetic field generated by the electric current. The proximity of the conductor and huge current density should lead to a significant magnetic field. The estimates or direct measurements have to be added to the paper to exclude that effect.

Reply:

Complying with the Reviewer's request, we have added on page 7 of the revised manuscript a discussion of the effects of the Oersted field of the current on the spectral shift. From COM-SOL simulations, we find that, for the used Pt electrode, the variation of the total magnetic field with the current is equal to 0.6 mT/mA. Calculations of the dispersion spectrum of magnons show that this variation leads to the shift of the frequency by about 0.01 GHz over the current range 1.3-2 mA, where the BEC peak is observed. This shift is smaller than the resolution of our measurements. Therefore, it cannot be seen in the experimental data. We note that the small influence of the Oersted field is due to the relatively small current density necessary to reach BEC conditions and due to the small effective magnetization of the system, where the anisotropy field is approximately equal to the saturation magnetization of the material.

The Reviewer writes:

In the manuscript, Authors present results only for one direction of the current through the Pt layer. However, to fully provide evidence of spin Hall effect driven injection, the results of opposite current polarity have to be presented. Otherwise, it is unclear whether the effect is not caused by the Spin Seebeck effect. The evidence must be presented and the corresponding discussion has to be added to the manuscript.

Reply:

We agree with the Reviewer that it is important to present a discussion of results for both current polarities. Complying with the Reviewer's request, we have added this discussion on pages 4 and 6 of the revised manuscript. We also added Supplementary Figure 1, which shows the BLS spectra for both polarities of the current. These data confirm that the onset of BEC can only be observed for positive currents, in agreement with the symmetry of the spin-Hall effect. We also included a figure showing the current dependence of the integral BLS intensity, which shows that the latter monotonically decreases with the increase of the magnitude of the negative current. This fact clearly indicates that the contribution of the Spin Seebeck effect, which is symmetric with respect to the change of the current polarity, is negligible in the studied system.

The Reviewer writes:

Could authors comment on the time scale of onset of the BEC in the observed current-driven scenario? Such discussion could greatly help the reader to understand the dynamics of the process.

Reply:

Following the Reviewer's recommendation, we have performed additional time-resolved measurements to determine the characteristic time scale of the BEC onset. We present the obtained results in Fig. 4c and discuss them on page 9 of the revised manuscript.

The Reviewer writes:

Concluding my report, I cannot recommend publication in its current form. However, I would be happy to reconsider my decision if the required evidence will be added to the manuscript.

Reply:

We hope that the Reviewer will find the added data and discussions sufficient and will recommend publication of the revised manuscript.

Response to Reviewer #3

The Reviewer writes:

This paper reports very interesting experiments showing evidences that it is possible to produce Bose-Einstein condensation (BEC) of magnons in a film of yttrium iron garnet (YIG) by spin currents generated by the conversion of dc electric currents through the inverse spin Hall effect (ISHE) in an adjacent platinum layer. In previous experiments on the formation of magnon BEC, the magnon system is excited by means of parametric pumping with pulsed microwave driving or rapid cooling by pulsed heating. In all experiments, the redistribution of the magnon system leading to the BEC is observed with Brillouin light scattering (BLS). In both cases the excitation of the magnon system raises the chemical potential, leading to the BEC when this reaches the minimum energy of the magnon dispersion relation in the YIG film, so in both processes the BEC state is observed by BLS only in a short time interval. Here, on the contrary, the excitation of the magnon system is made by a dc current, and the features of the BEC are observed with BLS in an equilibrium situation as the current intensity is raised above a critical value. In my opinion, the results presented in this manuscript constitute an important contribution to Bose-Einstein condensation phenomena which would be of interest to condensed matter physicists as well as to a broader community of scientists, and thus the paper could be published in Nature Communications. However, before acceptance of the manuscript, the authors would have to clarify some points and add some figures. The following specific points must be addressed:

Reply:

We thank the Reviewer for the positive evaluation of our work, for the appreciation of the importance of our findings, and for the recommendation to publish our manuscript after a revision. As discussed in detail below, we have added all the data requested by the Reviewer and provided all necessary clarifications. We hope that the Reviewer will find the revised manuscript suitable for publication in Nature Communications.

The Reviewer writes:

1- In lines 61-63, it is stated that "As a general rule of thumb, the nonlinear interactions between magnons is one of the main factors hindering the magnon BEC." However, as I understand, Ref. 3 shows that the magnon interactions are crucial for the magnon redistribution and also for the quantum coherence of the states with frequencies near the

minimum, which is a condition for a Bose-Einstein condensate, as shown in Ref. 23. Or am I wrong about the role of the magnon interactions?

Reply:

We agree with the Reviewer, that the role of magnon interactions in the evolution of the driven magnon gas is rather complex. This is mainly due to the fact that, contrary to the case of atomic gases, where the interaction is characterized by a constant parameter – the scattering amplitude, the amplitude of magnon-magnon interactions strongly depends on the wavevectors of interacting magnons. On one hand, nonlinear interactions are important for the efficient thermalization of injected magnons. They are particularly crucial in the case, if magnons are injected by the microwave pumping in a very small energy interval (case considered in Ref. 3). In this case, strong interactions are absolutely necessary to efficiently redistribute the injected magnons in the phase space. On the other hand, attractive interaction between magnons at the lowest-energy level results in the instability of the condensate. Therefore, the microwave-driven BEC can only be observed in a system, where the interactions are strong for all magnons except for those in the vicinity of the lowest-energy level (relatively thick YIG films).

In contrast, efficient driving of the magnon gas by the spin current requires that the magnetic film has nanometer-range thickness. In such films, the attractive interaction between lowest-energy magnons becomes very strong. This is the main reason why it was not possible to observe current-driven BEC up to now. In our work, by using the effects of the anisotropy, we minimize the magnon interactions for the lowest-energy magnons, which allows us to achieve formation of stable BEC. We note, however, that, for high-energy magnons, the interactions are still present (e.g., exchange-dominated interactions) enabling thermalization of the magnon gas. Additionally, in contrast to the microwave pumping, spin current creates magnons over very broad interval of energies, which makes the thermalization easier. In other words, this driving mechanism does not require very strong magnon interactions for the thermalization.

To address the Reviewer's comment, in the revised manuscript, we have modified the introduction to make our statements about nonlinear interactions more precise.

The Reviewer writes:

2- The authors state that the peak at 39 GHz corresponds to a perpendicular standing-wave (PSW) magnon mode and cite Refs 42,43. However, in both references the detected magnon modes have much smaller frequencies. Is there another reference that shows a quantitative estimate of the frequency of the PSW magnon for a 20 nm thick YIG film?

Reply:

First, we would like to note that the frequency of the observed mode coincides well with that predicted by the theory Ref. 42 (Fig. 1b). To our knowledge, our work is the first experimental observation of the PSW mode in such thin YIG films. In Ref. 11, the PSW mode with the frequency of 15 GHz was detected in somewhat thicker (70 nm) YIG. Addressing the Reviewer's comment, we have cited this paper together with Refs. 42,43, where the term "perpendicular standing-wave mode" was introduced.

The Reviewer writes:

3- If I understand correctly, the mechanism proposed for the formation of the BEC is the redistribution of the magnetic energy pumped into the PSW magnon by the spin current. Why couldn't the spin current excite magnons in a broad frequency and wave vector range, and these would then condense at the state with minimum frequency.

Reply:

The second statement of the Reviewer is correct. It is believed that the spin current excites magnons in a broad frequency and wave vector range, which then condense in the minimum-frequency state. However, it is not easy to prove this assertion experimentally, since there is no experimental technique, which allows one to directly measure the population of magnon states with large frequencies/wave vectors. In our work, we use the possibility to measure the population of the 39-GHz PSW mode to experimentally prove that the population of the high-frequency states is indeed affected by the spin current and that it saturates at the critical current, whereas the population of the lowest-energy state drastically increases, in agreement with predictions of Bose-Einstein statistics with a constant chemical potential. To make this clearer, we have modified the discussion on page 6 of the revised manuscript.

The Reviewer writes:

4- A crucial test for any experiment evidencing a new phenomenon involving magnons is the dependence on the magnetic field intensity. Figure 5 shows the field dependence of the critical current for BEC formation. The authors argue that the linear field dependence arises from the fact that the magnon relaxation rate varies linearly with frequency. However, this is true for magnons with small wave numbers, such as the one with minimum frequency. If I understand correctly, the magnons driven by the spin current are the ones in the PSW mode, and certainly these do not have relaxation rates that vary linearly with field.

Reply:

We agree with the Reviewer that the relaxation rate generally depends on the frequency and the wave vector of magnons and can show nonlinear dependence on the static magnetic field. The most common origin of this nonlinearity is associated with the effects of the dipolar interaction. In our system, dipolar effects are compensated by the anisotropy. Therefore, in the first approximation, one can assume a linear field dependence. We do not expect that this dependence should be ideally linear. However, as follows from the experimental data, an almost linear dependence can be expected. Complying with the Reviewer's criticism, we have modified the discussion of the data shown in Fig. 5 (page 10 in the revised manuscript).

The Reviewer writes:

5- Again, a crucial test for any experiment evidencing a new phenomenon involving magnons is the dependence on the magnetic field intensity. It is important that the authors include figures like Figs. 2a and 2b, showing the BLS spectra for another field value, say 200 mT, and one like Fig. 5 showing the field dependence of the BEC frequency.

Reply:

We thank the Reviewer for this comment. We agree that these additional data are necessary. Complying with the Reviewer's request, we have added in Fig. 5 the field dependence of the BEC frequency and discussed it on pages 9 and 10 of the revised manuscript. We have also provided figures like Figs. 2a and 2b for the largest field used in the experiment (Supplementary Fig. 2).

Reviewers' Comments:

Reviewer #1:

Remarks to the Author:

Unfortunately, the authors' arguments were unable to change my opinion on the manuscript.

From the physical point of view it is clear that the spin current creates excited spin states that can relax via magnons and thus can effectively pump magnons into the system. After "enough" magnons are created, one studies if/how they can condense. Previously, the O. Demokritov's group already experimentally demonstrated the possibility of such a condensation above a certain threshold using a different way of creating magnons in the system. How magnons are actually created in the system is the only major difference of present work with previous ones on the same subject.

Magnons interact with each other and this drives the system specific process of condensation. In the present manuscript the authors work with the Bi-substituted YIG films, characterized by stronger spin-orbit coupling and better magneto-optical properties than the original YIG. However, the interplay between demagnetizing and anisotropy effects is not universal – it cannot essentially modify the inter-magnon scattering processes everywhere. Moreover, by the nature of the shape-dependent demagnetizing field distribution (created by inhomogeneous dipolar fields) and system's anisotropy they cannot compensate each other in a way to render magnons non-interacting.

- > The paper mentioned by the Reviewer does not report similar
- > results. In this paper, which is the extension of the work
- > Ref. 11, a short-living BEC-like state was excited by the
- > strongly non-equilibrium rapid-cooling process, whereas the
- > weak spin-Hall effect could only result in a few-percent
- > variation of the onset threshold.

Even if the authors of the present manuscript call the observed in that paper state as "a short-living BEC-like state", it does not change the fact that everything the authors insist on in their Reply are just the details about the process of pumping magnons into the system and the details specific to the Bi-substituted YIG thin films. This all is important for experts in the field but such details do not require publication at the Nature level if the fact of room-temperature magnon BEC in thin YIG-like films itself has already been demonstrated in several earlier works.

To conclude, I cannot recommend publication of the manuscript in the Nature Communications. Instead, I would recommend to transfer it to the Communication Physics where authors could present more details important for experts in the field.

Reviewer #2:

Remarks to the Author:

After reviewing the revised version of the manuscript, I would like to recommend the article for publication in Nature Communications.

Reviewer #3:

Remarks to the Author:

As stated in my previous report, this paper reports very interesting experiments showing evidences that it is possible to produce Bose-Einstein condensation (BEC) of magnons in a film of yttrium iron garnet (YIG) by spin currents generated by the conversion of dc electric currents through the inverse spin Hall effect (ISHE) in an adjacent platinum layer.

In previous experiments on the formation of magnon BEC, the magnon system is excited by means of parametric pumping with pulsed microwave driving or rapid cooling by pulsed heating. In

all experiments, the redistribution of the magnon system leading to the BEC is observed with Brillouin light scattering (BLS). In both cases the excitation of the magnon system raises the chemical potential, leading to the BEC when this reaches the minimum energy of the magnon dispersion relation in the YIG film, so in both processes the BEC state is observed by BLS only in a short time interval. Here, on the contrary, the excitation of the magnon system is made by a dc current, and the features of the BEC are observed with BLS in an equilibrium situation as the current intensity is raised above a critical value.

In my opinion, the results presented in this manuscript constitute an important contribution to Bose-Einstein condensation phenomena which would be of interest to condensed matter physicists as well as to a broader community of scientists, and thus the paper could be published in Nature Communications.

I am happy with the explanations of the authors and the modifications made in the manuscript, and I recommend acceptance by Nature Communications.

Response to Reviewer #1

The Reviewer writes:

Unfortunately, the authors' arguments were unable to change my opinion on the manuscript.

From the physical point of view it is clear that the spin current creates excited spin states that can relax via magnons and thus can effectively pump magnons into the system. After "enough" magnons are created, one studies if/how they can condense. Previously, the O. Demokritov's group already experimentally demonstrated the possibility of such a condensation above a certain threshold using a different way of creating magnons in the system. How magnons are actually created in the system is the only major difference of present work with previous ones on the same subject.

Reply:

We respectfully disagree with the Reviewer's argument. As we already emphasized in our previous reply, the approach demonstrated in our current work enables creation of condensates, which are essentially different from those observed before. In contrast to microwave pumping, our approach allows one to create BEC without breaking the equilibrium between low-energy and high-energy magnons, which is known to result in extreme effective temperatures of the microwave-driven magnon gas (up to 5000 K). In addition, unlike the rapid-cooling process, our approach allows one to maintain stable state of the magnon condensate for unlimited time.

The Reviewer writes:

Magnons interact with each other and this drives the system specific process of condensation. In the present manuscript the authors work with the Bi-substituted YIG films, characterized by stronger spin-orbit coupling and better magneto-optical properties than the original YIG. However, the interplay between demagnetizing and anisotropy effects is not universal – it cannot essentially modify the inter-magnon scattering processes everywhere. Moreover, by the nature of the shape-dependent demagnetizing field distribution (created by inhomogeneous dipolar fields) and system's anisotropy they cannot compensate each other in a way to render magnons non-interacting.

Reply:

As we explained in detail in our previous reply, it is not our goal to "modify the inter-magnon scattering processes everywhere". To avoid instabilities and achieve stationary magnon condensate it is necessary to suppress the strong attractive interaction between lowest-energy magnons, which is typical for thin films. As clearly documented by the absence of the nonlinear frequency shift of the condensed magnons, this interaction is suppressed in Bi-substituted YIG films exhibiting perpendicular magnetic anisotropy.

The Reviewer writes:

**> The paper mentioned by the Reviewer does not report similar
> results. In this paper, which is the extension of the work
> Ref. 11, a short-living BEC-like state was excited by the
> strongly non-equilibrium rapid-cooling process, whereas the
> weak spin-Hall effect could only result in a few-percent
> variation of the onset threshold.**

Even if the authors of the present manuscript call the observed in that paper state as "a short-living BEC-like state", it does not change the fact that everything the authors insist on in their Reply are just the details about the process of pumping magnons into the

system and the details specific to the Bi-substituted YIG thin films. This all is important for experts in the field but such details do not require publication at the Nature level if the fact of room-temperature magnon BEC in thin YIG-like films itself has already been demonstrated in several earlier works.

Reply:

We would like to stress once more that, until now, no stable stationary BEC has been demonstrated in thin films. In our opinion, such observation opens new avenues for studies of magnon BEC, enables implementation of a large variety of exciting new experiments, and paves the way for implementation of integrated microscopic quantum magnonic and spintronic devices.

The Reviewer writes:

To conclude, I cannot recommend publication of the manuscript in the Nature Communications. Instead, I would recommend to transfer it to the Communication Physics where authors could present more details important for experts in the field.

Reply:

In our opinion, our results fully meet the criteria for publication in Nature Communications. They “will be important to the field and advance understanding in a way that will move the field forward”.

Response to Reviewer #2

The Reviewer writes:

After reviewing the revised version of the manuscript, I would like to recommend the article for publication in Nature Communications.

Reply:

We thank the Reviewer for the recommendation to publish the revised manuscript in Nature Communications.

Response to Reviewer #3

The Reviewer writes:

As stated in my previous report, this paper reports very interesting experiments showing evidences that it is possible to produce Bose-Einstein condensation (BEC) of magnons in a film of yttrium iron garnet (YIG) by spin currents generated by the conversion of dc electric currents through the inverse spin Hall effect (ISHE) in an adjacent platinum layer.

In previous experiments on the formation of magnon BEC, the magnon system is excited by means of parametric pumping with pulsed microwave driving or rapid cooling by pulsed heating. In all experiments, the redistribution of the magnon system leading to the BEC is observed with Brillouin light scattering (BLS). In both cases the excitation of the magnon system raises the chemical potential, leading to the BEC when this reaches the minimum energy of the magnon dispersion relation in the YIG film, so in both processes the BEC state is observed by BLS only in a short time interval. Here, on the contrary, the excitation of the magnon system is made by a dc current, and the features of

the BEC are observed with BLS in an equilibrium situation as the current intensity is raised above a critical value.

In my opinion, the results presented in this manuscript constitute an important contribution to Bose-Einstein condensation phenomena which would be of interest to condensed matter physicists as well as to a broader community of scientists, and thus the paper could be published in Nature Communications.

I am happy with the explanations of the authors and the modifications made in the manuscript, and I recommend acceptance by Nature Communications.

Reply:

We thank the Reviewer for the appreciation of the importance of our findings and for the recommendation to publish our manuscript in Nature Communications.